# Effects of Diets Enriched with Conventional or High-Oleic Canola Oils on Vascular Endothelial Function: A Sub-Study of the Canola Oil Multi-Centre Intervention Trial 2 (COMIT-2), a Randomized Crossover Controlled Feeding Study

**DOI:** 10.3390/nu14163404

**Published:** 2022-08-18

**Authors:** Kristin M. Davis, Kristina S. Petersen, Kate J. Bowen, Peter J. H. Jones, Carla G. Taylor, Peter Zahradka, Karen Letourneau, Danielle Perera, Angela Wilson, Paul R. Wagner, Penny M. Kris-Etherton, Sheila G. West

**Affiliations:** 1Department of Biobehavioral Health, Pennsylvania State University, State College, PA 16802, USA; 2Department of Nutritional Sciences, Pennsylvania State University, State College, PA 16802, USA; 3Department of Nutritional Sciences, Texas Tech University, Lubbock, TX 79409, USA; 4Richardson Centre for Functional Foods and Nutraceuticals, University of Manitoba, Winnipeg, MB R3T 6C5, Canada; 5Department of Food and Human Nutritional Sciences, University of Manitoba, Winnipeg, MB R3T 2N2, Canada; 6Department of Physiology and Pathophysiology, University of Manitoba, Winnipeg, MB R3E 3P5, Canada; 7The Canadian Centre for Agri-Food Research in Health and Medicine, St. Boniface Hospital, Winnipeg, MB R2H 2A6, Canada

**Keywords:** flow-mediated dilation, conventional canola oil, high-oleic canola oil, cardiovascular disease risk

## Abstract

Partial replacement of saturated fatty acids (SFA) with unsaturated fatty acids is recommended to reduce cardiovascular disease (CVD) risk. Monounsaturated fatty acids (MUFA), including oleic acid, are associated with lower CVD risk. Measurement of flow-mediated dilation of the brachial artery (FMD) is the gold standard for measuring endothelial function and predicts CVD risk. This study examined the effect of partially replacing SFA with MUFA from conventional canola oil and high-oleic acid canola oil on FMD. Participants (*n* = 31) with an elevated waist circumference plus ≥1 additional metabolic syndrome criterion completed FMD measures as part of the Canola Oil Multi-Centre Intervention Trial 2 (COMIT-2), a multi-center, double-blind, three-period crossover, controlled feeding randomized trial. Diet periods were 6 weeks, separated by ≥4-week washouts. Experimental diets were provided during all feeding periods. Diets only differed by the fatty acid profile of the oils: canola oil (CO; 17.5% energy from MUFA, 9.2% polyunsaturated fatty acids (PUFA), 6.6% SFA), high-oleic acid canola oil (HOCO; 19.1% MUFA, 7.0% PUFA, 6.4% SFA), and a control oil blend (CON; 11% MUFA, 10% PUFA, 12% SFA). Multilevel models were used to examine the effect of the diets on FMD. No significant between-diet differences were observed for average brachial artery diameter (CO: 6.70 ± 0.15 mm, HOCO: 6.57 ± 0.15 mm, CON: 6.73 ± 0.14 mm; *p* = 0.72), peak brachial artery diameter (CO: 7.11 ± 0.15 mm, HOCO: 7.02 ± 0.15 mm, CON: 6.41 ± 0.48 mm; *p* = 0.80), or FMD (CO: 6.32 ± 0.51%, HOCO: 6.96 ± 0.49%, CON: 6.41 ± 0.48%; *p* = 0.81). Partial replacement of SFA with MUFA from CO and HOCO had no effect on FMD in participants with or at risk of metabolic syndrome.

## 1. Introduction

Metabolic syndrome (MetS) is a growing public health concern, affecting 34% of adults in the United States [1]. While various definitions have been used in research [2], MetS is a cluster of cardiometabolic risk factors including abdominal obesity, dyslipidemia, elevated fasting glucose, and hypertension [3]. MetS is associated with higher risk of developing atherosclerosis [4,5,6,7,8] and coronary artery disease [2], as well as increased risk of cardiovascular disease (CVD) morbidity and mortality [9,10,11,12,13]. Endothelial dysfunction is common in MetS and is implicated in the pathogenesis of atherosclerosis and CVD [14,15,16,17,18]. A cornerstone of CVD risk reduction is partial replacement of dietary saturated fatty acids (SFA) and trans fatty acids (TFA) with unsaturated fatty acids, both monounsaturated fatty acids (MUFA) and polyunsaturated fatty acids (PUFA). Partial replacement of SFA with MUFA, plant-derived MUFA in particular, reduces CVD risk [19,20,21,22] largely by improving lipid/lipoprotein profiles [23,24]. Similarly, diets high in MUFA from plant-based foods have been shown to reduce risk of MetS [25]. However, the effect of oleic acid, the predominate MUFA, on endothelial function remains unclear [26,27,28,29].

The vascular endothelium is integral in the maintenance of vascular health [30]. The gold standard for non-invasive measurement of endothelial function is brachial artery flow-mediated dilation (FMD) [31]. Lower (i.e., worse) FMD is associated with higher CVD risk factor burden [32,33], and increased risk of incident CVD events [34,35,36,37]. All components of MetS negatively impact endothelial function [38]; consequently, improving MetS components may lead to improvements in endothelial function and, thus, in long-term cardiovascular health outcomes.

Canola oil and high-oleic canola oil (HOCO), key dietary sources of oleic acid, have well-established lipid-lowering effects [20,24]. However, less research has examined the effect of partially replacing SFA with MUFA, particularly high-oleic oils, on endothelial function. Consumption of a meal high in oleic acid impaired postprandial FMD in healthy men [39], but improved markers of endothelial function in men with hypertriglyceridemia [40]. Additionally, past research has found that dietary patterns high in oleic acid improve endothelial function and reduce CVD risk in individuals with type 2 diabetes compared to a diet high in PUFA [41], while others have shown no benefits of diets high in oleic acid on endothelial function in individuals with insulin resistance [42]. Since trans fats were removed from the US Generally Recognized As Safe list [43], high oleic oils are increasingly being used to replace trans fats in commercial food products [44]. Consequently, consumption of high-oleic oils is expected to increase [44]. Additional research is needed to clarify the health effects of increasing oleic acid consumption, especially in individuals with or at risk of MetS.

The first Canola Oil Multi-Centre Intervention Trial (COMIT-1; NCT01233778), a five-period randomized crossover controlled feeding study, compared the health effects of five different oils (canola oil, high oleic acid + docosahexaenoic acid canola oil, high oleic acid canola oil, flax oil, and safflower oil) in individuals with elevated cardiometabolic risk. Results indicated that canola oil and HOCO were associated with a significant reduction in abdominal fat mass [45]. COMIT-2 (NCT02029833) was then designed to probe the promising results observed with the high-oleic oils in COMIT-1. Details of the study protocol and primary outcomes have been published previously [24,46]. Briefly, a three-period, randomized, crossover, controlled feeding study was conducted to examine the effect of canola oil, HOCO, and a control oil blend designed to match the fatty acid composition of the average American diet. The primary outcome for the COMIT-2 trial was visceral and abdominal adiposity; however, other markers of cardiometabolic health, including FMD, were included as secondary outcomes.

The aim of the present sub-study of COMIT-2 is to examine the effect of canola oil and HOCO on FMD compared to the control oil. We hypothesized that partial replacement of SFA with oleic acid would improve FMD, but that canola and HOCO effects would not differ significantly because of their similar high oleic acid content. 

## 2. Materials and Methods

COMIT-2 was a randomized controlled trial, with a 3-period crossover, double-blind, controlled feeding design [24,46]. FMD was a secondary outcome in this trial and was only measured in a sub-sample of individuals from two study centers: St. Boniface Hospital Albrechtsen Research Centre (SBRC, Winnipeg, MB, Canada) and The Pennsylvania State University (PSU, State College, PA, USA). Diet periods were 6 weeks long, separated by ≥4-week washout periods. Diet orders were counterbalanced to limit order effects, with six sequences possible. For each participant, the odds of being assigned to any one of the diet sequences were equal. The study was approved by each university’s respective ethics boards (Ethical approval numbers: University of Manitoba Biomedical Ethics Board, HS18178; St. Boniface Hospital Research Review Committee, RRC/2014/1377; Pennsylvania State University IRB, CR00003924), and all procedures were followed in accordance with the Declaration of Helsinki. Written informed consent was obtained from each participant at screening, prior to study enrollment. This trial was registered at clinicaltrials.gov (NCT02029833).

### 2.1. Participants

A total of 53 participants completed the study at PSU and SBRC. Eligible participants were between 20 and 65 years of age and had an elevated waist circumference (≥94 cm for men and ≥80 cm for women), plus at least one other International Diabetes Federation MetS criterion [3], including fasting blood glucose ≥5.6 mmol/L; triglycerides ≥1.7 mmol/L; high-density lipoprotein (HDL)-cholesterol <1 mmol/L (men) or <1.3 mmol/L (women); and blood pressure ≥130 mmHg (systolic) and/or ≥85 mmHg (diastolic). Individuals with diabetes mellitus, or with thyroid, liver, or kidney disease were ineligible to participate. Current smokers, and those consuming >1 (females) or >2 (males) alcoholic drinks per day were also ineligible. Individuals taking medications that affect lipid metabolism or endothelial function were excluded. Due to the effect of the menstrual cycle on FMD [47,48], pre-menopausal women (*n* = 21) were excluded from participating in FMD testing. Thus, a subset of participants (*n* = 32) underwent FMD measurement at baseline and at the end of each diet period; see Figure 1 for the participant flow chart. Participants who only completed one FMD test, as well as FMD scans that were not scorable, were excluded, resulting in a final sample of 31 men and post-menopausal women.

### 2.2. Intervention

Across all diet periods, participants consumed a fixed basal diet designed for weight maintenance. Macronutrient composition of the diets, including the oils, is presented in Table 1. The Harris–Benedict Formula [49] was used to estimate participants’ calorie needs, with calorie levels increasing in 300 kcal increments. If a substantial change in weight was seen during the first two weeks of the first diet period, adjustments to calorie level were made. After this initial adjustment period, participants’ calorie level was maintained for the duration of the study. 

The basal diet was supplemented with three different oils (one for each diet period), which provided 18% of total energy each day (e.g., 60 g per 3000 kcal). Diets included a conventional canola oil diet (17.5% MUFA, 9.2% PUFA, 6.6% SFA), a high-stability/high-oleic canola oil blend diet (HOCO; 19.1% MUFA, 7.0% PUFA, 6.4% SFA), and a control diet with an oil blend containing butter oil/ghee, flaxseed oil, safflower oil, and coconut oil (11% MUFA, 11% PUFA, and 13% SFA). Canola oil and HOCO contain similar levels of SFA. Due to the higher MUFA content of HOCO, this diet was lower in PUFA and higher in MUFA relative to the CO diet. The fatty acid composition of the control oil blend was designed to approximate the average fatty acid intake of US adults as reported in NHANES (2015–2016): 12% MUFA, 8% PUFA, and 12% SFA [50]. Oils were blended into smoothies made with frozen strawberries, orange sherbet, and milk. By design, all foods provided were identical across the three diet periods, except for the oils incorporated in the smoothies. 

### 2.3. Outcome Assessment

On two consecutive days at baseline and at the end of each diet period, participants reported to the clinical research center at the respective site following a 12 h fast from food or drink (besides water) and a 48 h abstinence from alcohol. Fasting blood samples were taken on both days at each time point. Weight, waist circumference, and seated blood pressure were measured on just one of the two days at each time point. FMD was measured at the first pre-trial baseline visit and at the end of each diet period, for a total of four measurements. Height was measured at screening only.

### 2.4. Endothelial Function

FMD was measured in accordance with established guidelines [31]. To reduce variability, a single sonographer performed all FMD measurements at each site. The sonographers at both sites were American Registry for Diagnostic Medical Sonography (ARDMS)-certified in vascular sonography and received additional training in flow-mediated dilation scanning prior to beginning the study. Three electrocardiogram (EKG) electrodes were applied in a modified lead II configuration to measure heart rate. An occlusion cuff was applied to the participant’s right forearm, with the arm extended at approximately a 45° angle away from the body. Participants rested quietly in a supine position for 10 min prior to beginning the FMD procedure. The sonographer then placed an ultrasound probe on the participant’s upper arm to image the brachial artery. Once a clear image was obtained, continuous images were acquired via Realtime 2-D B-Mode Ultrasound Imaging and Doppler, using a GE Logiq e duplex ultrasound scanner with a 12L linear array broadband transducer (General Electric Medical Systems, Chicago, IL, USA). Imaging was performed at 50% acoustic output power with a center frequency of 10.0 MHz. Doppler evaluation of blood flow was performed at 5.0 MHz center frequency at acoustic output power of 70%. Digital video capture was performed at a rate of 5 frames/s for 8 min. Baseline Doppler was performed for a duration of 10 s, followed by 50 s of imaging to obtain the baseline brachial artery diameter (BAD). Cuff inflation was initiated at 1 min into the recording, at a cuff pressure of 250 mmHg, using a Hokanson Rapid Cuff Inflation System (AG101 cuff inflator air source with E20 rapid cuff inflator; Hokanson, Bellevue, WA, USA). Cuff deflation was performed at the end of five minutes of inflation; an additional two minutes of video were captured post cuff-release. A ten-second Doppler sample was obtained immediately post deflation, returning to 2D imaging within 15 s of deflation. 

Automated edge detection software (Brachial Analyzer for Research Version 6; Medical Imaging Applications, LLC, Iowa City, IA, USA) was used to continuously measure BAD, defined as the distance from the anterior to the posterior arterial media. Frames in which less than 30% of the region of interest could be accurately measured were rejected. A pre-occlusion average baseline diameter was obtained from the 50 s of images taken prior to cuff inflation. Peak BAD was defined as the first occurrence of the largest diameter value observed after the occlusion cuff was released. FMD, expressed as percentage change from baseline, was calculated as follows: FMD = [(peak BAD − baseline average BAD)/baseline average BAD] × 100(1)

Each scan was scored by two independent scorers. If the FMD value obtained by these scorers differed by ≥2 percentage points, a third scorer was included. The average of the two most similar scores was used in the statistical analyses. When all three scores differed by ≥2 percentage points, the scan was considered not scorable, and was excluded from analysis. For 10 s at both the beginning of baseline and immediately after cuff release, duplex-pulsed Doppler was used to measure blood flow velocity. Flow (mL/min) was calculated as follows: velocity time integral × cross-sectional area of the vessel (π × (baseline average BAD/2)2) × heart rate. Reactive hyperemia (RH), the change in flow after cuff release, was calculated: (peak flow−baseline flow)/baseline flow × 100.

### 2.5. Blood Sample Collection and Analysis

On two consecutive days at baseline and at the end of each diet period, 12 h fasting blood samples were collected. Ethylenediaminetetraacetic acid plasma tubes were centrifuged immediately post-collection. Serum in separator tubes was allowed to clot for 30 min, and then separated by centrifugation. Samples were aliquoted into cryovials, and stored in a locked freezer at −80 °C. Frozen samples were then packed in coolers with dry ice, and shipped to St. Michael’s Hospital (Toronto, ON, Canada), where blood samples from all study sites were analyzed. 

### 2.6. Statistical Methods

Statistical analyses were performed using SAS (version 9.4; SAS Institute, Cary, NC, USA). Variables were assessed for normality prior to beginning analyses; non-normally distributed variables are presented as geometric means with geometric standard deviations. Preliminary analyses used Welch’s *t*-tests to test for significant differences between the FMD subsample and the overall study sample at baseline. Primary analyses examined between-diet mean differences in FMD, assessing the main effect of diet using multilevel models. Secondary analyses used multilevel models to examine change in FMD from baseline, assessing the main effect of diet. The mixed models procedure (PROC MIXED) was used for all analyses, with the diet period (Canola, HOCO, or control) nested within subjects and the reference group set to the control diet. Significance was accepted at *p* < 0.05. When a significant main effect was found, the Tukey–Kramer method was used to compare diet periods, adjusting for multiple pairwise comparisons. To ensure good model fit, Pearson’s Chi Squared Likelihood Ratio Tests and the associated *p*-values were used to compare covariance matrix structures; variance components were the best fitting covariance matrix structure. The univariate procedure (PROC UNIVARIATE) was used to assess normality of the residuals, specifically considering Q–Q plots and the Kolmogorov–Smirnov test, with a *p*-value of <0.05 indicating a non-normal distribution. Variables with non-normally distributed residuals were log-transformed as needed to yield a more normal shape. Multilevel models were also used following the same methods to check for order effects using the diet sequence by diet interaction and for carryover effects using the visit number by diet interaction, for between-center differences in the main effects of diet on FMD using the study center by diet interaction, and to test for between-diet differences in baseline BAD, baseline blood flow, peak blood flow, and RH. Past research indicates that a sample size of 30 provides 90% power to detect a 25% relative change in FMD [51].

## 3. Results

Participant characteristics at baseline are presented in Table 2; additional participant characteristics can be found in Appendix A Appendix A. For the subset of participants that completed FMD testing, between four and six participants were assigned to each diet sequence. At baseline, mean FMD was 6.48 ± 0.49%. In this sample, 72.4% of participants (*n* = 21) met the full criteria for MetS at baseline. The geometric mean waist circumference was 106.9 ± 2.8 cm for males, and 103.5 ± 2.4 cm for females. The percentage of participants who met each individual MetS criterion at baseline ranged from 31.0% for fasting glucose to 96.6% for waist circumference. Only diastolic blood pressure differed significantly between the FMD subsample and the overall COMIT-2 sample (*p* < 0.001) (Table 3). Analyses did not indicate the presence of order effects (*p* = 0.43) or carryover effects (*p* = 0.59). Study center (*p* value = 0.38) and sex (*p* = 0.86) were not significant predictors of FMD and were, thus, dropped from subsequent models. 

No significant between-diet differences in FMD (*p* = 0.81), baseline BAD (*p* = 0.72), peak BAD (*p* = 0.80), baseline blood flow (*p* = 0.72), peak blood flow (*p* = 0.29), or RH (*p* = 0.41) were observed. Additionally, there were no significant effects of MetS status (*p* = 0.94) or the interaction of MetS status and diet on FMD (*p* = 0.49) in the endpoint-to-endpoint analysis; results from the most parsimonious endpoint-to-endpoint models are presented in Table 3. Change-from-baseline analysis also revealed no significant effect of diet period on change in FMD from baseline (*p* = 0.62).

## 4. Discussion

This study examined the effects of partially replacing SFA with MUFA from canola oils on endothelial function. The present results do not support our hypothesis that partially replacing SFA with oleic acid from canola oils for 6 weeks would improve FMD in adults with or at risk of metabolic syndrome. 

Previous research supports a beneficial effect of olive oil, naturally high in oleic acid, on FMD. A meta-analysis of eight randomized controlled studies showed that olive oil interventions resulted in a 0.76% increase in FMD [52]. However, it is unclear whether oleic acid is responsible for this effect, as olive oil contains other potentially mediating compounds, including polyphenols [53]. At least one previous study has reported beneficial effects of oleic acid, specifically, on FMD. Ryan and colleagues (2000) compared the effects of a diet high in linoleic acid (a PUFA) with a diet high in oleic acid on FMD in a small (*n* = 11) sample of males with type 2 diabetes, and found that the oleic-acid-rich diet resulted in a 2.2% increase in FMD [41]. There are several possible explanations for the difference in findings. First, Ryan and colleagues used 2-month-long diet periods, slightly longer than the 6-week periods used in the present study. It is plausible that the extra time was necessary for oleic acid to induce beneficial effects on endothelial function. Additionally, participants in Ryan and colleagues’ study all had type 2 diabetes, while participants in the present study were generally healthy but at risk of developing MetS; it is plausible that the beneficial effects of oleic acid on FMD may differ in individuals with and without diabetes. The methodology of the study by Ryan and colleagues also differs substantially from the present study. COMIT-2 was a controlled feeding study in which participants were provided all meals and snacks throughout each diet period. In contrast, Ryan and colleagues provided a single session of nutritional counseling with a registered dietitian prior to starting each diet period. While it is unclear what the participants consumed, specifically, the authors provide evidence that the participants changed their dietary patterns by measuring the oleic and linoleic acid content of adipocyte membranes in participants after each diet period. Finally, Ryan and colleagues assessed FMD of the femoral artery rather than the brachial artery; past research suggests that the two measures are not correlated [54], and as such this may be a factor in the different results obtained.

A growing evidence base suggests that oleic acid intake may either not significantly impact FMD, as found in the present study, or perhaps may even impair FMD [39,42,55,56]. For example, Berry and colleagues (2008) conducted a randomized crossover trial with 17 male participants and found that a meal high in oleic acid (from high-oleic sunflower oil) decreased postprandial FMD by 3% compared to baseline, while a meal high in stearic acid (an SFA; from a shea butter blend) did not significantly change postprandial FMD [39]. Vogel and colleagues (2000) similarly found that a meal high in oleic acid from olive oil significantly impaired postprandial FMD (by 31%) in 10 healthy participants, using a randomized crossover design. This study also included a meal supplemented with canola oil instead of olive oil; in this condition, no significant change in postprandial FMD was observed. Interestingly, Vogel and colleagues also tested two meals that added a salad with balsamic vinegar or vitamins E and C, and found that these additions attenuated the decrease in FMD by 65% and 71%, respectively, suggesting that the impairment in FMD after oleic acid consumption may be driven by oxidative stress [55]. Moving to habitual diet-based paradigms, Sanders and colleagues (2013) found no effect of partially replacing SFA with MUFA (~5% replacement) for 24 weeks in a parallel arm randomized controlled trial with 121 insulin-resistant men and women [42]. Participants were provided with oils, fatty spreads, and mayonnaise to help achieve the assigned fatty acid composition, and were provided dietary instructions, but were otherwise free to consume their habitual diet. In COMIT-2, slightly more SFA was replaced with MUFA (conventional canola: 6.4%; HOCO: 6.6%); however, the amount of replacement is small enough to be feasible for the general population. To our knowledge, COMIT-2 is the first controlled feeding study to directly assess the impact of partially replacing SFA with MUFA on FMD. 

The study is limited by its small sample size; because FMD has a high degree of within-person variability, this study may have been underpowered to detect treatment effects. While a sample size of 30 is considered to provide 90% power to detect a 25% relative change in FMD [51], this is the first study to examine the effect of partially replacing SFA with MUFA on FMD in the context of a controlled feeding study. Thus, data were lacking for power calculations and the true statistical power of the sample is difficult to determine. Still, use of a crossover design increases the statistical power of the sample [57]. Smaller crossover studies examining FMD have found significant effects with 11 [41,58], 19 [59], and 20 [60] participants and only two experimental conditions. It is therefore plausible that the present sample of 31 participants across three diet periods was adequately powered. Substantial racial homogeneity (79.3% white) further limits this study; it is unclear whether results would generalize to other racial or ethnic groups. Additionally, the participants had relatively high average FMD at baseline (6.48 ± 0.49%), indicating good vascular health despite the presence of multiple MetS risk factors. In contrast, Ryan and colleagues (2000) observed a mean FMD of 3.90% in individuals with type 2 diabetes in a linoleic-acid-rich diet condition, and a significantly higher 6.12% FMD with an oleic-acid-rich diet [41]. It is plausible that, in COMIT-2 participants, damage to the vasculature was not yet manifested in a way that could be meaningfully improved via short-term dietary interventions. Additional research is warranted to investigate the effects of partially replacing SFA with oleic acid in individuals with more advanced endothelial dysfunction. Further, the basal diet provided during all three study periods was also intentionally designed to be a healthy, weight-maintenance diet [24]. While the controlled feeding design is a strength of the present study, it is possible that the relatively high FMD we observed in the control diet could be related to the healthy basal diet. Whether replacing SFA with MUFA in the context of an average Western dietary pattern would improve FMD remains unclear.

## 5. Conclusions

The present study did not demonstrate improvements in FMD with partial replacement of dietary SFA with dietary oleic acid from canola oils in individuals with or at risk of MetS. While the cardiovascular benefits of partially replacing SFA with MUFA are well-established [19,20,21,22,23,24,25], questions remain about underlying mechanisms. Based on the increasing rate of oleic acid consumption [44], longitudinal prospective studies are needed to evaluate the replacement of a larger percentage of dietary SFA with oleic acid on CVD risk, as well as longer duration clinical studies to further examine the health effects of high-oleic oil consumption. Beyond this, given that plant-based oleic acid is a component of liquid oils that contain a milieu of other compounds, including fatty acids, polyphenols and antioxidants, there is a need to clarify how each affects CVD risk. This information could explain the endpoint discrepancies in the high-oleic oil literature and, thus, better inform future dietary guidance about which liquid oils best reduce CVD risk.

## Figures and Tables

**Figure 1 nutrients-14-03404-f001:**
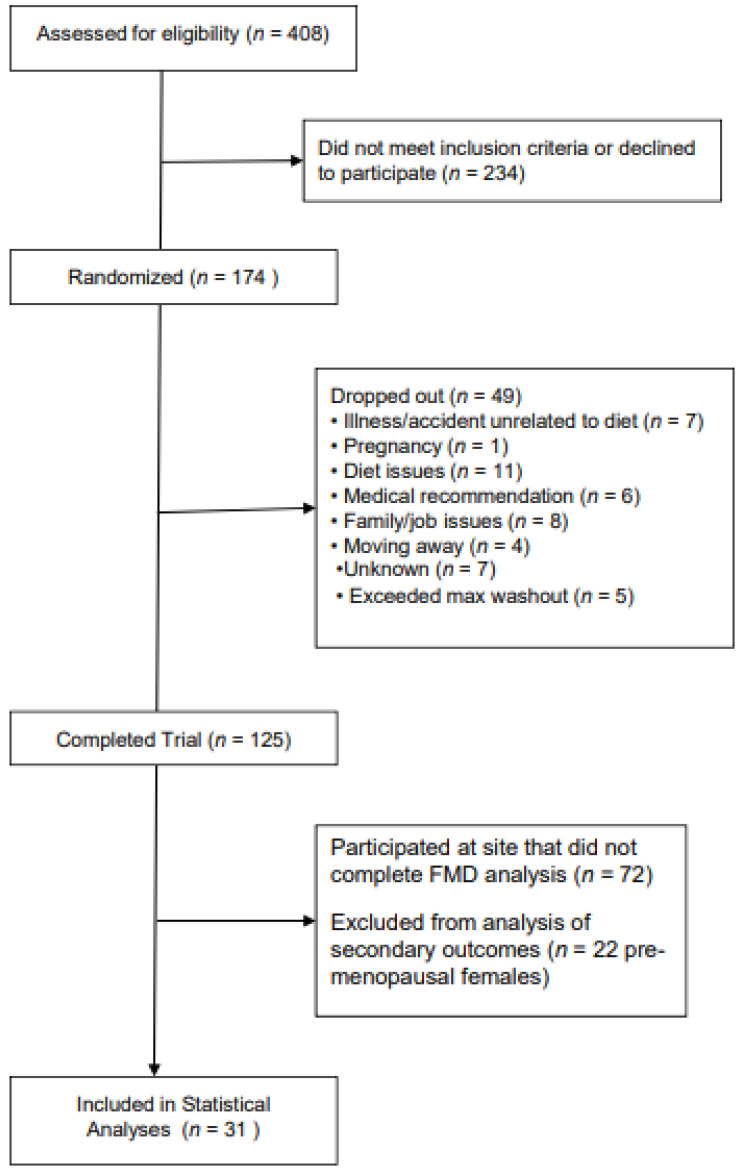
CONSORT participant flow diagram of the COMIT-2 participants for inclusion in the flow-mediated dilation analyses. COMIT, Canola Oil Multi-Centre Intervention Trial; max, maximum; FMD, flow-mediated dilation.

**Table 1 nutrients-14-03404-t001:** Macronutrient composition of the three experimental diets, including oils ^1,2^.

	Canola	HOCO	Control
Carbohydrate	50.79	50.79	50.75
Protein	15.87	15.87	15.71
Fat	35.26	35.26	35.21
MUFA	17.45	19.11	10.50
Oleic Acid	15.55	17.86	5.92
PUFA	9.21	7.02	9.96
α-linolenic acid	2.10	0.76	1.73
Linoleic Acid	6.42	5.56	7.28
SFA	6.56	6.43	12.26

^1^ Average composition of the 7-day rotating menu; est. at the 3000 kcal level using Food Processor Nutrition Analysis Software (ESHA Research; Salem, OR, USA); ^2^ All values presented as percentage of total energy. MUFA, monounsaturated fatty acids; PUFA, polyunsaturated fatty acids; SFA, saturated fatty acids; HOCO, high-oleic acid canola oil.

**Table 2 nutrients-14-03404-t002:** Participant characteristics at baseline.

		Starting Diet		
Variable	Overall *n* = 31	Canola *n* = 11	HOCO *n* = 11	Control *n* = 9
Sex	*n* (%)			
Male	20 (67)	9 (82)	7 (64)	4 (44)
Female	11 (33)	2 (18)	4 (36)	5 (56)
Anthropometric Measures	Mean ± SD			
Age (years)	43 ± 14	39 ± 14	43 ± 13	45 ± 14
BMI (kg/m^2^)	32.1 ± 5.6	35.4 ± 7.0	30.0 ± 2.2	31.5 ± 5.5
MetS Criteria				
Waist Circumference (cm)	Male: 106.9 ± 1.1 ^1^ Female: 103.5 ± 1.1 ^1^	115.2 ± 1.1 ^1^ 111.2 ± 1.0 ^1^	99.9 ± 1.1 ^1^ 97.4 ± 1.1 ^1^	101.5 ± 1.1 ^1^ 105.6 ± 1.0 ^1^
TG (mmol/L)	1.6 ± 0.8	1.7 ± 0.7	1.7 ± 1.1	1.2 ± 0.6
HDL-C (mmol/L)	Male 1.1 ± 0.3	1.2 ± 0.4	1.1 ± 0.3	1.2 ± 0.4
	Female 1.6 ± 0.4	1.4 ± 0.2	1.6 ± 0.6	1.7 ± 0.5
SBP (mmHg)	125 ± 13	125 ± 7	120 ± 15	128 ± 16
DBP (mmHg)	86 ± 9	86 ± 8	84 ± 11	86 ± 9
Fasting Glucose (mmol/L)	5.3 ± 0.5	5.4 ± 0.5	5.2 ± 0.5	5.4 ± 0.5
Met full MetS criteria *	*n* = 21 (72.4%)	9 (81.8%)	5 (50%)	7 (77.8%)
Additional CVD risk factors				
Total Cholesterol (mmol/L)	5.1 ± 1.0	5.1 ± 1.2	5.4 ± 0.9	5.0 ± 0.7
LDL-C (mmol/L)	3.2 ± 0.8	3.1 ± 0.9	3.4 ± 0.9	3.0 ± 0.6

BMI, body mass index; SD, standard deviation; MetS, metabolic syndrome; TG, triglycerides; SBP, systolic blood pressure; DBP, diastolic blood pressure; CVD, cardiovascular disease; LDL-C, low-density lipoprotein cholesterol; HDL-C, high-density lipoprotein cholesterol; ^1^ geometric means and geometric standard deviations presented due to non-normally distributed variable. * To meet International Diabetes Federation definition of MetS, a person must have an elevated waist circumference, plus at least 2 additional MetS criteria, including elevated triglycerides, low HDL-C, high blood pressure, and elevated fasting glucose.

**Table 3 nutrients-14-03404-t003:** Results of multilevel models.

	Baseline	Control	Canola	HOCO	*p*
Avg BAD (mm)	6.55 ± 0.14	6.73 ± 0.14	6.70 ± 0.15	6.57 ± 0.15	0.72
Peak BAD (mm)	6.97 ± 0.14	7.14 ± 0.14	7.11 ± 0.15	7.02 ± 0.15	0.80
FMD (% change)	6.48 ± 0.49	6.41 ± 0.48	6.32 ± 0.51	6.96 ± 0.49	0.81
Baseline Flow ^#^	176.7 ± 19.1	241.0 ± 24.9	246.5 ± 26.4	246.5 ± 26.4	0.72
Peak Flow	1067.8 ± 93.0	1168.4 ± 89.2	1376.6 ± 99.7	1307.9 ± 92.3	0.29
RH	503.9 ± 47.8	393.53 ± 46.1	351.1 ± 47.9	376.4 ± 47.9	0.41
FMD Change from Baseline (mm)	N/A	−0.2 ± 0.6	0.1 ± 0.7	0.7 ± 0.6	0.62

^#^ non-normally distributed variables are presented as geometric means; *p* values from multilevel models conducted using mixed models procedure (PROC MIXED); Avg, average; BAD, brachial artery diameter; FMD, flow-mediated dilation; RH, reactive hyperemia; N/A, not applicable.

## Data Availability

Available upon request.

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
