# Peer review of "Effects of Diets Enriched with Conventional or High-Oleic Canola Oils on Vascular Endothelial Function: A Sub-Study of the Canola Oil Multi-Centre Intervention Trial 2 (COMIT-2), a Randomized Crossover Controlled Feeding Study"

_nutrients, 2022, doi:10.3390/nu14163404_

Round 1
Reviewer 1 Report
Abstract
The abstract is very confusing and it is not reflecting the main document. Please rework on it.
Introduction
-Describe the definition of metabolic syndrome from the below article (https://www.mdpi.com/1660-4601/18/4/1773)
- I am not convinced with the sentences present between the lane numbers of 69-73.
-Discuss a sentence about COMIT-2 in the introduction. How authors can directly jump into aim?
-Please rewrite the aim
-Ethical approval number is missing.
-Write accurately about inclusion and exclusion criteria of this study.
-Methodology and results were well written but the authors need to incorporate the volume of blood collected in EDTA and plain vacutainer.
-In Discussion, please add the global and meta-analysis studies to compare the current study results.
-It seems this manuscript can be considered as short communication rather than original article.
Reviewer 2 Report
A well-respected group of investigators with an established track-record in these types of studies, which are not easy to execute, and frequently provide limited information.. This is basically a pilot study to pump out another publication (perhaps a thesis).! Not a problem (we've all been there!), but I do not find anything novel in the study - nor any major findings. The results - no effect - are highly likely attributed to the sample size. In addition to the stats, what is the physiological/biological relevance. Despite the title, this is NOT the COMIT study as you are reporting from merely 2 centers and only 31 subjects (9-11 per intervention). I strongly suggest that you omit reference to the COMIT study in the title as well as in your rationale. Presumably the study was not powered from the standpoint of measuring FMD. As such the extensive discussion bringing in olive oil, MUFA in general and dietary advice for the general population is really not warranted. The Discussion should focus on the limited results that were obtained. I commend the authors for acknowledging some of these concerns in their Limitations section.
Given all the above, the last few lines (353-355) are really the crux, well known by everyone, but have nothing to do with the content or findings from this paper
The authors are some of the leading exponents in the field and have the ability to shape scientific opinion and thought. Accordingly, this paper needs to be presented from some other standpoint..
Round 2
Reviewer 2 Report
My comments have
been addressed